# Assessment of Preload Loss after Cyclic Loading in the OT Bridge System in an “All-on-Four” Rehabilitation Model in the Absence of One and Two Prosthesis Screws

**DOI:** 10.3390/ma15041582

**Published:** 2022-02-20

**Authors:** Mario Cesare Pozzan, Francesco Grande, Edoardo Mochi Zamperoli, Fabio Tesini, Massimo Carossa, Santo Catapano

**Affiliations:** 1Department of Prosthodontics, University of Ferrara, Via Luigi Borsari 46, 44121 Ferrara, Italy; mariocesare.pozzan@edu.unife.it (M.C.P.); mochizamperolie@gmail.com (E.M.Z.); fabio.tesini@edu.unife.it (F.T.); cts@unife.it (S.C.); 2Department of Surgical Sciences CIR Dental School, University of Turin, Via Nizza 230, 10126 Turin, Italy; massimo.carossa@unito.it

**Keywords:** preload loss, OT bridge, prosthetic connection, implant-supported prosthesis, loosening torque, all-on-four

## Abstract

The aim of this study was to evaluate the stability of prosthetic screws after applying cyclic loadings in an “all-on-four” rehabilitation model with the OT Bridge system. The model was tested both with and without anterior screws. Four implant analogues following the “all-on-four” concept were inserted in an edentulous mandibular resin model. An OT Bridge system with a Cr–Co prosthetic framework was fabricated. Depending on the presence or absence of one or two anterior screws on the implant analogues, three groups were created, i.e., Gr.1: three tightening screws, Gr.2: two tightening screws, Control Group: four tightening screws. Each single group underwent subsequent 400,000 cyclic loads, simulating approximately a year of chewing by using a dynamometer machine. This cycle was repeated five times for each group, and preload loss values were evaluated on each prosthetic screw after each cycle. All the data obtained were analyzed by one-way ANOVA and Student’s *t*-test. No statistically significant differences after intragroup analysis were found. A statistically significant difference within the Gr.1 between the screws in positions 33 and 36, equal to 15.2% (*p*-value = 0.0176), was found. The OT Bridge seems a useful system to maintain the retention of a prosthesis during mechanical stress conditions even in the absence of one screw in an “all-on-four” rehabilitation. This could represent a good solution to solve the esthetic problem of the screw buccal access hole for fixed solutions.

## 1. Introduction

One of the fixed solutions that has been commonly used for several years in implant prosthodontics for the complete-arch rehabilitation of the upper and lower jaws is the “all-on-four” protocol [1,2,3]. This technique consists in the insertion in the mandibular ridge of two anterior intra-foraminal implants perpendicular to the occlusal plane and two posterior angulated implants emerging in the second premolar/first molar position [4]. For the upper jaw, the position could be the same with respect to the anatomy of the maxillary sinuses. In this way, the “all-on-four” allows a complete arch rehabilitation using a fixed solution with only four implants, achieving a favorable implant distance with a short cantilever [5]. The most frequent complication, described in the literature for screw-in systems, is the loss of preload that is defined as the force established between the screw turns and the abutment thread [6,7]. Many factors could influence the loss of preload, which is a prodromal manifestation of the connecting screw fracture [8,9,10]. The relationship between preload and tightening torque (force exerted to tighten a screw) depends on several factors: the type of implant–prosthetic connection, the morphology and material of the abutment, the design and material of the screw, tolerances between screw and thread, the morphology of the implant fixture, the type of prosthetic rehabilitation, iatrogenic errors [10,11].

In order to connect implants with prosthetic frameworks, the most used solution is today represented by the Multi Unit Abutment (M.U.A) which has the function to change the implant connection from an internal hexagon to an external one. When titled implants are inserted to correct the implant axis and to solidarize distal implants with the anterior ones, angulated M.U.A components are necessary. In this system, the prosthetic screw is the only component that allows the binding between the prosthetic framework and the implants [12,13,14,15].

Recently, another screw-retained system, called OT Bridge^®^ (Rhein 83 srl, Bologna, Italy), has been introduced on the dental market as an alternative to M.U.A. [16]. The OT Bridge is composed of a low-profile attachment for overdenture, the OT Equator, a sub-equatorial component represented by an interchangeable acetal ring called Seeger, and a cylindrical titanium “extragrade” abutment. This abutment is cemented in the prosthetic framework and, at its retentive extremity, is provided of a cavity designed for the insertion of the acetal ring which occupies the undercut created, allowing a first retention of the prosthesis with the OT Equator. In this way, the Seeger ring provides a secure and functioning elastic retention system that guarantees the stable housing of the prosthesis on the OT equator attachment even in the absence of the prosthetic screw which has a diameter of 1.6 mm.

An important clinical issue, arising with “screw-retained” rehabilitations, especially in the case of severely resorbed jaws, is the presence of a buccal screw access channel that constitutes an anesthetic problem independently of the prosthetic material use [17,18,19,20,21]. This issue could be managed with a “cemented-retained” solution, although the advantages of the screw-retained prosthesis are lost. Another way to solve this problem could be using angulated M.U.A abutment only for less than a 30° disparallelism [22]. In OT Bridge rehabilitations, this esthetic problem is maintained or amplified because the OT Equator follows the implant axis.

A possible solution, retaining the benefits of a screw-in prosthesis, may be provided by the OT-Bridge system [16] in the absence of one or two anterior prosthetic screws [23,24]. In this way, the stability of the OT Bridge system is provided by both the posterior screws and the interlocking system between the acetal Seeger, inserted in the extragrade abutment, and the subequatorial region of the OT equator attachments in the anterior region. However, it is not clear and there are no data available in the literature about the stability of this system during and after cyclic loadings without one or two screws.

Therefore, the aim of this study was to evaluate, in an “all-on-four” model with an OT Bridge prosthetic system, the stability of its connecting screws after cyclic loadings, both in the presence and in the absence of the anterior screws. The null hypotheses (H0) considered for this study were the absence of removal torque difference between the model without one or two screws and the model with all four screws (intergroups analysis—H0 nr.1) and the absence of removal torque difference between the values of the screws within each aforesaid solution (intragroup analysis—H0 nr.2).

## 2. Materials and Methods

### 2.1. Samples Preparation

A resin model, already described in a previous article [16], was fabricated (Figure 1a,b). Four implant analogues (3.5 × 10 mm; NobelBiocare, Kloten, Switzerland) with internal hexagonal connections were placed in intraforaminal position following the “all-on-four” concept [4]. Each implant was placed according to a predetermined angulation: the axes of implants in canine/lateral incisor position were orthogonal to the occlusal plan, while those of implants in second premolar/first molar area were distally tilted by 30°. A calibrated hole was performed in the center of the model to fix it to the machine for the dynamometric control of the loads.

A milled cobalt–chrome framework for hybrid prosthesis connecting the four implant analogues was fabricated from a digital STL file. Extragrade-titanium angulated and straight abutments (Rhein 83, Bologna, Italy) were used for the model.

According to the most predicted protocols described in the literature, two monolithic zirconia crowns, whose central fossa was 7 mm distal to the last implant analogue emergences (36 and 46), were distally positioned and cemented on the cobalt–chrome framework using the dual resinous cement OT Cem (Rhein83, Bologna, Italy) [25,26]. The cementation phase included the insertion of the crown on the appropriate housing and, to optimize the fit between the crown and the framework, the placement of a load of 10 N for 5 min through MTS Acumen (MTS Systems S.R.L, Turin, Italy) [27]. The load was then applied on the cemented monolithic zirconia crowns.

According to the indications proposed by the manufacturer (Rhein83, Bologna, Italy), the connection screws were tightened at 25 Ncm by means of the torque-controlled dynamometric micromotor Implant-Med-Plus (W&H, Brusaporto—BG, Italy); for this study, the white acetalic Seeger (standard seal) was used. After 10 min, a second screwing was performed at 25 Ncm, according to the screwing protocol proposed by Winkler et al. [28].

Depending on the presence or absence of one or two anterior screws on the implant analogues, three groups of five screws for removal torque evaluation after cyclic loadings were created and compared to each other (Figure 2):Group 1 (Gr.1): Group with 3 tightening screws (33, 36, 46) inserted on 4 implant analogs without the screw in position 43.Group 2 (Gr.2): Group with 2 tightening screws (36, 46) inserted on 4 implant analogs without the anterior screws 33 and 43.Control Group (Gr.CTR): Group with 4 tightening screws (33, 43, 36, 46 inserted) on 4 implant analogs. This group was the same as that a previous study by Catapano et al. [16].

### 2.2. Cyclic Test

The MTS Acumen, which is a dynamometer machine, was used to approximately simulate the developed effort of a year of chewing loads [29,30]. Each loading test consisted of 400,000 subsequent cycles with a variable force from 40 N to 400 N at the frequency of 1.6 Hz, orthogonally to the occlusal plane. The vertical excursion of the piston was 0.16 mm. (Figure 3). This cycle was repeated five times for each group.

To simulate the oral cavity environment during cyclic load, the model was placed in a thermostatic water bath at a constant temperature of 37 °C. (Figure 4). To evaluate preload loss values, after each single cyclic load, the registration of the connection screw torque removal was performed using Implant-Med-Plus^®^ on each tightening screw. After each single unscrewing session, the implant–prosthetic connection screws and the acetalic Seegers inserted in the cobalt–chrome framework were changed.

### 2.3. Statistical Analysis

All the data obtained in this in vitro study were analyzed by statistical evaluations, which were carried out through the GraphPad Prism data analysis program^®^ 9.2.0 and through statistical tests for variance, i.e., one-way ANOVA and Student’s *t*-test.

For both tests, the statistical significance was set at 0.05 (*p*-value < 0.05).

Using one-way ANOVA with Bonferroni correction, the following data were evaluated:differences between the averages mean values of preload loss for Gr.1, Gr.2, and Gr.CTR;differences in preload loss percentage values of the screws in positions 36 and 46 between the three different groups;differences in preload loss percentage values of the screws inside each group;

The following statistical analyses were performed using Student’s *t*-test:differences in percentage torque loss of screws positioned in zone 33 for Gr.1 and Gr.CTR.differences in percentage torque loss between the screws positioned in zones 36 and 46 in Gr.2.

The null hypotheses considered were:absence of removal torque difference between the three groups (intergroups analysis—H0 nr.1);absence of removal torque difference between the values of the screws within each group (intragroup analysis—H0 nr.2).

## 3. Results

The raw data of unscrewing for each connecting screw after cyclic loading were used to extrapolate the individual percentage values of preload loss used for statistical analysis. The results are shown in Table 1:

From one-way ANOVA with Bonferroni correction, the following results were obtained:There were no significant statistical differences in torque mean values between Gr.1, Gr.2, and Gr.CTR (*p*-value = 0.2670).There were no significant statistical differences between the percentages of preload loss for the screw in position 36 between the three different groups (*p*-value = 0.0569).There were no significant statistical differences between the percentages in preload loss for the screw in position 46 between the three different groups (*p*-value = 0.4177).The difference between the screws in positions 36 and 46 and 33 and 46 in Gr.1 showed no statistically significant difference, with *p*-value = 0.0193.A statistically significant difference between the percentages of preload loss between the screws in positions 33 and 36 in Gr.1 was found equal to 15.2% (*p*-value = 0.0176) (Table 2).There was no statistical significant difference between the percentages of torque loss between screws in positions 33 and 46 (*p*-value = 0.4184) and between screws in positions 36 and 46 (*p*-value = 0.3123) in Gr.1.

Student’s *t*-test indicated that:There were no statistically significant differences between Gr.1 and Gr.CTR for the screw in position 33 (*p*-value = 0.1114).A *p*-value = 0.6465 was calculated, and therefore there was no statistical significant difference between the individual screws 36 and 46 within Gr.2.

## 4. Discussion

Abutments used for implant-supported prosthodontics have been largely studied, and nowadays the most used solutions are the screw-retained abutments. The use of this type of abutments and their mechanical complications, like the loss of preload and the fracture of prosthetic screws, are well known [31]. The stability of the OT Bridge system is provided by both the screw and the interlocking system between the acetalic Seeger in the extragrade abutment and the subequatorial region of the OT equator attachment. Only few works have been carried on this system out, and research and predictable protocols are only at the beginning, but the clinical results seem to be promising. However, studies on the follow-up period and the number of cases reported with “all-on-four” rehabilitation using the OT bridge system are limited [22,23,24]. No studies have been proposed to evaluate the influence of removing one or two anterior screws in an “all-on-four” rehabilitation and the stability of this specific system after mechanical loading. This condition can be considered a realistic clinical case where, for esthetic issues, the clinician may choose to avoid the screw hole on one or two anterior implants. In this way, the system is more stressed because the tensile force, resulting from the applied posterior loadings, is counteracted only by the retention capacity of the Seeger. Furthermore, in this study we chose positions 36 and 46 to apply the cyclic loads in order to examine the worst-case condition of loading (on the distal cantilever).

Regarding the results of this study, no statistically differences in mean torque loss values were found between the groups tested, so the null hypothesis (H0 nr.1) was accepted. On the other hand, the null hypothesis for the intra-groups evaluation (H0 nr.2) was not accepted because of the statistically significant difference within Gr.1 between the prosthetic screws in positions 33 and 36.

The absence of statistically significant differences between Gr.1, Gr.2, and Gr.CTR seems to indicate that the OT bridge system could passivate the cyclic loading forces without compromising the entire implant–prosthetic connection even in case of higher mechanical stress than in the normal configuration, such as that occurring in the absence of one tightening screw. Similar results were obtained by Cervino et al. [24] who confirmed the stability of the OT Bridge system in the absence of one screw in an “all-on-four” rehabilitation in a FEM analysis.

From the comparison of the values of Gr.1 (three screws inserted), a statistically significant difference could be noted between the screws in positions 33 and 36 but not between the screws in positions 33 and 46 and 36 and 46 (Table 2).

For the distal 36 and 46 screws, where the loading cycle forces were vertically dissipated (in compression), the OT Bridge system appeared to guarantee stability, because a significant preload loss was not observed even when one or two anterior prosthetic screws were removed. There were also no differences between the two distal screws in the same model with or without the mesial homolateral screw.

In the model with three screws (Gr.1), the screw in position 33 was more stressed by the loading cycles. Probably, the tensile force, exerted by the lever arm of the distal cantilever in positions 36 and 46, acting in the opposite direction to that of the preloading force of the screw, induced more stress on screw 33. This fact could not be appreciated in the Control Group because of the presence of the second anterior screw (43) that divided the lever arm into additional parts compared to Gr.1. Considering these values, the OT Bridge system seems to be able to dissipate the force even in situations where the mechanical stress should result in a greater overload (model with three screws instead of four).

In this way, it seemed that the Seeger maintained the prosthesis stable and fixed through its retentive action even in the absence of its anterior connection screws (43, 33), as observed for a system with all four screws in place (Gr.CTR).

Theoretically, the removal of one or two connecting screws should lead to an overload on the remaining screws and therefore a loss of preload which, from the analysis conducted, did not occur. This could probably depend on the “Seeger action” which was kept in place, thus being able to carry out its retentive action on the abutment even in the absence of the connection screw.

In support of this thesis, the system was forced beyond the recommended limits by removing the two front screws (33, 43) in Gr.2. This is not a recommended clinical condition, but through the tests we performed, no statistically significant differences for the preload loss between Gr.2 and Gr.CTR were found. In this way, by keeping in place the acetalic Seeger also in the absence of one or two screws, the OT Bridge retentive capacity seems to hinder the loss of preload forces.

The limits of this study are principally related to its in vitro nature because the resin model tested did not allow us to understand and determine the effect on soft and hard peri-implant tissues and the number of five unscrewing sessions per group resulted in a limited amount of data. The model used for this study was based on an edentulous lower arch, but for the upper jaws the results could be different because of the different circle arch described and the different implant angulations [22,31]. On the other hand, the bilateral forces exerted on the cantilever to simulate stressful conditions in the oral cavity may be unrealistic, because the chewing function foresees different loading conditions and is multidirectional, not vector-specific. The bilateral cyclic load was performed to test the entire system in a more stressful condition. Another important limit is the lack of data on the Seeger deformation and the holding capacity of the Seeger after cyclic loadings. This could be examined in further research. Additional in vitro, in vivo, and clinical trials with long-term follow-up, larger samples and different conditions are therefore required to better understand the feasibility of the OT Bridge system.

## 5. Conclusions

In conclusion, from the results of this in vitro study, the OT Bridge seems to be a stable system when using three screws in the “all-on-four” technique applied to a mandibular model. This in vitro finding could suggest a good way to solve the esthetic problem of the screw buccal access hole for “screw-retained” fixed solutions, maintaining the advantages of these types of rehabilitations. The mean torque loss values were not statistically different for the groups tested.

The retentive function mediated by the Seeger is useful to maintain the retention of a prosthesis even in mechanical stress conditions. However, different loading conditions, number of cyclic loads, frequency and positions of the applied loads have to be evaluated in the future in order to better understand the OT Bridge system stability.

## Figures and Tables

**Figure 1 materials-15-01582-f001:**
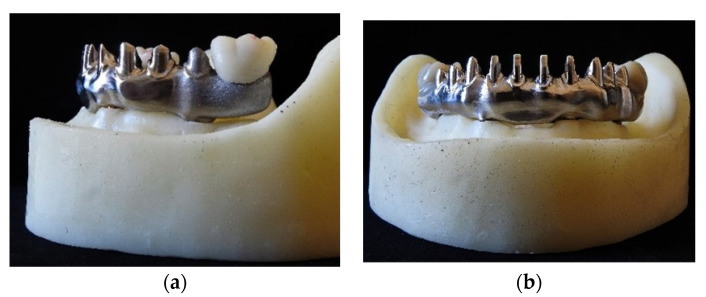
Lateral (**a**) and frontal views (**b**) of the model used.

**Figure 2 materials-15-01582-f002:**
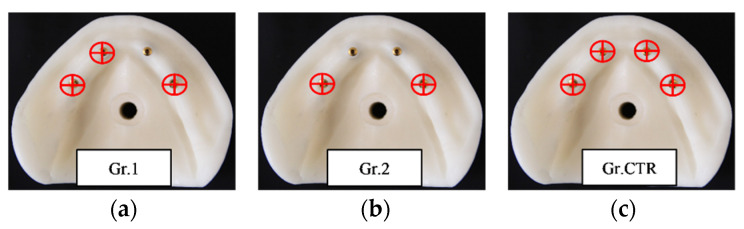
Representative images of the screws’ placement in the three different groups. From the left to the right: Gr.1 (**a**), Gr.2 (**b**), and Gr.CTR (**c**).

**Figure 3 materials-15-01582-f003:**
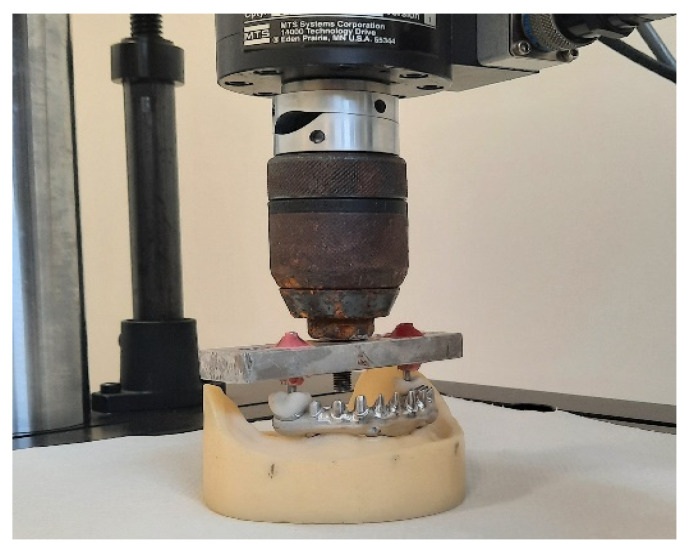
Representative image of the model placement under the MTS Acumen.

**Figure 4 materials-15-01582-f004:**
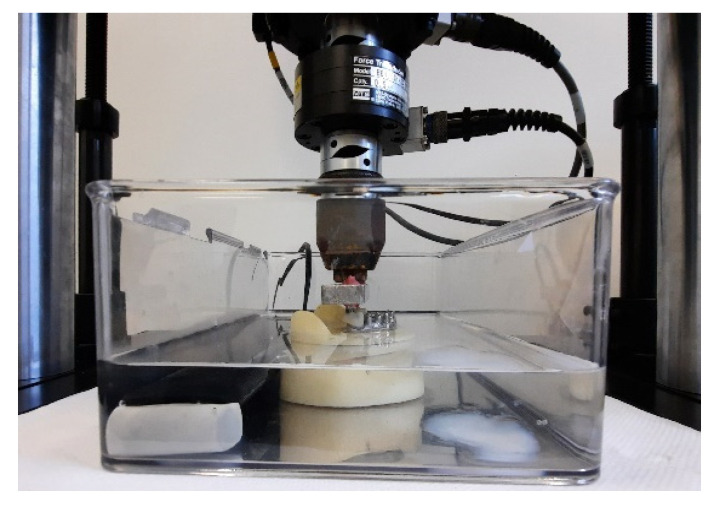
Image of the model during a cyclic load in the water bath.

**Table 1 materials-15-01582-t001:** Unscrewing values and percentages of preload loss obtained for each group.

	**GR.CTR** **Screwed at 25 Ncm**				**GR.1** **Screwed at 25 Ncm**			**GR.2** **Screwed at 25 Ncm**	
	33	43	36	46	33	36	46	36	46
First unscrewing	16 Ncm = 36%	14 Ncm = 44%	13 Ncm = 48%	12 Ncm = 52%	12 Ncm = 52%	15 Ncm = 40%	15 Ncm = 40%	13 Ncm = 48%	12 Ncm = 52%
Second unscrewing	15 Ncm = 40%	14 Ncm = 44%	13 Ncm = 48%	13 Ncm = 48%	18 Ncm = 28%	18 Ncm = 28%	15 Ncm = 40%	15 Ncm = 40%	12 Ncm = 52%
Third unscrewing	14 Ncm = 44%	14 Ncm = 44%	15 Ncm = 40%	13 Ncm = 48%	13 Ncm = 48%	16 Ncm = 36%	15 Ncm = 40%	18 Ncm = 28%	16 Ncm = 36%
Fourth unscrewing	17 Ncm = 32%	15 Ncm = 40%	11 Ncm = 56%	16 Ncm = 36%	11 Ncm = 56%	18 Ncm = 28%	15 Ncm = 40%	16 Ncm = 36%	16 Ncm = 36%
Second unscrewing	16 Ncm = 36%	10 Ncm = 60%	15 Ncm = 40%	16 Ncm = 36%	12 Ncm = 52%	18 Ncm = 28%	15 Ncm = 40%	10 Ncm = 60%	12 Ncm = 52%
SD	1.14 Ncm = 4.56%	1.95 Ncm = 7.80%	1.67 Ncm = 6.69%	1.87 Ncm = 7.48%	2.77 Ncm = 11.10%	1.41 Ncm = 5.66%	0	2.52 Ncm = 12.20%	2.31 Ncm = 8.76%
Mean	15.6 Ncm = 37.6%	13.4 Ncm = 46.4%	13.4 Ncm = 46.4%	14 Ncm = 44%	13.2 Ncm = 47.2%	17 Ncm = 32%	15 Ncm = 40%	14.4 Ncm = 42.4%	13.6 Ncm = 45.6%

**Table 2 materials-15-01582-t002:** One way ANOVA test for GR.1. In yellow, are the results of the statistic comparisons between the screws in positions 33 and 36.

Bonferroni’s Multiple Comparisons Test	Mean Diff	95.00% CI of Diff	Below Threshold?	Summary	Adjusted *p* Value	
33 vs. 36	15.20	2.556 to 27.84	Yes	*	0.0176	A–B
33 vs. 46	7.200	−5.444 to 19.84	No	ns	0.4184	A–C
36 vs. 46	−8.000	−20.64 to 4.644	No	ns	0.3123	B–C
Test details	Mean 1	Mean 2	Mean Diff	SE of diff	n1	n2
33 vs. 36	47.20	32.00	15.20	4.549	5	5
33 vs. 46	47.20	40.00	7.200	4.549	5	5
36 vs. 46	32.00	40.00	−8.000	4.549	5	5

## Data Availability

Not applicable.

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
