# Peer review of "Assessment of Preload Loss after Cyclic Loading in the OT Bridge System in an “All-on-Four” Rehabilitation Model in the Absence of One and Two Prosthesis Screws"

_materials, 2022, doi:10.3390/ma15041582_

Round 1
Reviewer 1 Report
In this study, the authors evaluated the stability of prosthetic screws after cyclic loadings, with and without anterior screws. A system with a Cr-Co prosthetic framework was used which was based on the presence or absence of one or two anterior screws. Every group of systems were cyclically loaded by using a dynamometer machine. This load condition is adequate to simulate chewing, but authors should explain choosing points of loading and absence screws positions.
Why weren't there included a combination of absence screws under loading points 36 and 46? or loading points at 33 and 43?
These different groups would give added results of each prosthetic screw after each cycle. All additional data could show differences for the intragroup. Therefore, please particularly discuss these issues - boundary conditions in the article.
The article contains some errors. Figure 1 includes two images so, please correct the description under figure: In the firth and in the third image (second instead of third). Also, please use the same abbreviations for the name of groups. In line 190 should be capital letters of gr.2.
The article can be accepted after minor revision according to the above notices (corrections methodological description and minor text editing).
Author Response
Dear reviewer,
as you suggested:
- we explain the choosing points of loading and the positions of absence screws in line 210-216. Also in the conclusion we specify that future studies with different loading conditions have to be done in order to better understand the system stability
- we corrected the legends under the images
- We changed Gr.2 in capital letter
Thank you for your support
Reviewer 2 Report
The aim of the paper is to evaluate the stability of prosthetic screws after cyclic loadings in a model which simulate an “All- on-four” rehabilitation with OT- Bridge system, with and without anterior screws. The title corresponds to the manuscript content. The manuscript is well structured and the adequate research methods and statistical analysis software are used. All tables, figures and references are cited in the text. There are no missing references in the list.
Remrks
- The English of the abstract and the whole manuscript should be checked, especially for sequence of tenses.
Abstract
- The sentence on rows 2-23: “The results show no 22 statistically difference for the intragroup analysis were found.” Needs revision.
Materials and methods
- I would reckommend this chapter to be devided into 3 sub-sections: 1) Samples preparation, 2) Cyclic test and 3) Statistical analysis. Thus, the methods used in the research will be more clearly presented.
- Fig. 1 on p.3:
- Show the designations -a) and b) on the pictures.
- Check the figure caption:”... In the firth and in the third image it could...” May be there is an error - there are only 2 pictures.
- P.3, row 131: „... [29,230] Each loading test...” May be reference 30?
Results
- Table 1. The first column is not clear enough. Is it unscrewing at 1o, 2o up to 5o (degree) or may be this means 1st unscrewing? Please check.
Conclusion
- Actually, there is no Conclusion chapter. The main finding of the research should be highlited in the separate Conclusion chapter.
Author Response
Dear reviewer,
as you suggested:
- we checked the english for the whole manuscript and corrected it
- we made a subdivision of materials and methods
- we corrected the legends under the images and the reference error
- we clarify the first column of Table 1 in the results
- we added a conclusion paragraph
Thank you for your support